# Lid loop-mediated proton transfer revealed in the Fe/αKG-dependent decarboxylase TraH
Xuehua Zheng[1,2,7], Rui Ge[3,7], Zhiyong Guo [4,7], Mathias Girbig [5,6], Johannes Freitag [1], Aitao Li[4], Georg K. A. Hochberg[5,6], Shu-Ming Li [3] ✉, Gert Bange [1,5] ✉ & Liujuan Zheng [1,5] ✉

TraH, a fungal decarboxylase from *Penicillium crustosum*, belongs to the isopenicillin N synthase (IPNS) subfamily of non-heme iron/α-ketoglutarate (αKG)-dependent enzymes. IPNS enzymes are characterized by N- and C-terminal insertions that can reshape the active site. However, the functional and mechanistic roles of these elements, particularly in fungal decarboxylases, remain largely unexplored. Here, we report crystal structures of TraH in complex with various substrates, revealing a N-terminal loop, serving as lid and undergoes substrate-dependent dynamic conformational rearrangements. Upon binding of crustosic acid, this lid loop forms a hydrogen-bonding network to stabilize a water molecule, which mediates interaction between the conserved K191 residue in the DSBH (double-stranded β-helix) core and the substrate. Mutagenesis and QM/MM metadynamics simulations suggest that proton transfer from K191 is mediated by water molecules stabilized by the lid loop, supporting a decarboxylation mechanism distinct from the canonical strategy of direct carboxylate stabilization by DBSH core-located basic residues in other αKG-dependent decarboxylases. Interestingly, this loop is dispensable for the desaturation of crustosic acid methyl ester, thereby pinpointing its essential role to the precise positioning of the carboxylate substrate for proton transfer during decarboxylation. Evolutionary and structural analyses reveal significant variation in lid loop composition across IPNS enzymes, indicating its contribution to substrate recognition and functional diversification. Overall, our findings uncover a regulatory element in iron/ αKG-dependent enzymes and offer insights into how non-core structural elements contribute to catalytic mechanism and evolution.

Mononuclear nonheme iron/α-ketoglutarate dioxygenases (αKGDs) are a diverse family of enzymes that catalyze a range of oxidative reactions fundamental to microbial metabolism, including the biosynthesis of pharmaceutically important secondary metabolites[1,2]. While hydroxylation remains the most studied reaction, research has expanded to reveal the mechanistic diversity and biological importance of other αKGD-catalyzed processes, such as oxidative decarboxylation[3–8]. A prominent example is the iron/αKG decarboxylase TraH from *Penicillium crustosum*, a fungus known for

producing mycotoxins and food spoilage[9]. TraH is involved in the biosynthesis of terrestric acid, a secondary metabolite suggested to possess antibiotic, antibacterial, and antiviral activity due to its structural features[10]. Utilizing αKG as a co-substrate, TraH catalyzes the oxidative decarboxylation of crustosic acid (**1a**), resulting in the formation of an exocyclic double bond in an intermediate[10], as depicted in Fig. 1A. More recently, Zhu et al. identified ThnC, a homolog of TraH with 44% sequence identity, from the fungus *Trichoderma harzianum*[11]. ThnC is involved in the biosynthesis

[1]Philipps-Universität Marburg, Center for Synthetic Microbiology (SYNMIKRO) & Department of Chemistry, Marburg, Germany. [2]Guangzhou Municipal and Guangdong Provincial Key Laboratory of Molecular Target & Clinical Pharmacology, the NMPA and State Key Laboratory of Respiratory Disease, School of Pharmaceutical Sciences, Guangzhou Medical University, Guangzhou, China. [3]Philipps-Universität Marburg, Fachbereich Pharmazie, Institut für Pharmazeutische Biologie und Biotechnologie, Marburg, Germany. [4]State Key Laboratory of Biocatalysis and Enzyme Engineering, Hubei Key Laboratory of Industrial Biotechnology, School of Life Sciences, Hubei University, Wuhan, China. [5]Max-Planck Institute for Terrestrial Microbiology, Marburg, Germany. [6]Philipps-Universität Marburg, Department of Biology, Marburg, Germany. [7]These authors contributed equally: Xuehua Zheng, Rui Ge, Zhiyong Guo. ✉e-mail: shuming.li@staff.uni-marburg.de; gert.bange@synmikro.uni-marburg.de; liujuan.zheng@mpi-marburg.mpg.de; zhengli@staff.uni-marburg.de

**Fig. 1 | Reactions catalyzed by fungal αKG-dependent decarboxylases and structural features of IPNS subfamily. A** Decarboxylation reactions catalyzed by the iron/αKG-dependent decarboxylases TraH and ThnC. The carboxyl groups removed during catalysis are highlighted in red. **B** Structural features of the IPNS subfamily. The representative crystal structure of IPNS (PDB ID: 1BK0) highlight the conserved double-stranded β-helix (DSBH) core in green, and the extended N-terminal and C-terminal insertions are colored tv-red and tv-blue.

of trihazone D and catalyzes the decarboxylation of a substrate containing tetronate scaffold, which is also present in crustosic acid, but with different side chain.

Structurally, αKGDs share a conserved double-stranded β-helix (DSBH) fold that coordinates the catalytic iron and defines the geometry of the active site, providing a framework for substrate recognition and catalysis[12,13]. Variations in surrounding insertions and loops diversify this scaffold, giving rise to distinct subfamilies[14]. Within this context, iso-penicillin N synthase (IPNS)-like enzymes form a subfamily including TraH and ThnC, distinguished by extended N- and C-terminal insertions that reshape the substrate-binding site (Fig. 1B)[14]. These structural extensions are implicated in substrate binding across various IPNS members[14–19]. For example, structural studies have shown that N-terminal segments of varying lengths create distinct active-site environments in enzymes such as IPNS, T7H, TropC, and EFE[15–19]. However, the influence of such analogous insertions on TraH and ThnC remains largely unexplored, and systematic investigations into their structure and mechanism across the subfamily are therefore warranted.

In this study, we present the crystal structures of TraH in complex with different substrates, revealing that a flexible lid region, derived from an N-terminal insertion, adopts distinct conformations upon binding substrates for decarboxylation and desaturation. Structural analysis, mutagenesis, and combined molecular dynamics (MD) and quantum mechanics/molecular mechanics (QM/MM) metadynamics simulations revealed an uncharacterized mechanism where the lid loop mediates a hydrogen-bonding network to stabilize water molecules, facilitating proton transfer during oxidative decarboxylation of crustosic acid. Interestingly, this hydrogen-bonding network is not essential for TraH catalyzed desaturation of crustosic acid methyl ester. Additionally, our evolutionary analysis of lid loop conservation and variability among TraH homologs, combined with structural analysis of IPNS enzymes, offers valuable insights into the structure-function relationships and catalytic specificity within this subfamily. These findings highlight the adaptive functional role of the lid loop in the IPNS family, revealing an interesting structural adaptation in αKG dioxygenases and providing insight into their catalytic mechanisms.

## Results
### Crystal structure of TraH highlights a flexible lid loop stabilized by ligand binding
To facilitate structural elucidation, we co-crystallized TraH with Mn²⁺ (commonly used to substitute iron and prevent substrate catalysis during the crystallization of Fe/αKG-dependent enzymes), αKG, and crustosic acid (**1a**). The obtained structure was refined to a resolution of 2.4 Å (Table S1), with two protein molecules present in each asymmetric unit

(Fig. 2A, PDB ID: 9IG5, Supplementary Data 1). As shown in Fig. 2B and S1C, the overall structure of TraH aligns well with the defining features of the IPNS subfamily, centered around a DSBH core with extended N- and C-terminal insertions. The DSBH core (Fig. 2B, green) consists of eight antiparallel β-strands organized into two layers: the major sheet (βI, βVIII, βIII, and βVI) and the minor sheet (βII, βVII, βIV, and βV). The N-terminal insertion extends the DSBH core with two β-strands positioned at each end of the major sheet (β2–5), and an additional strand (β1) aligned parallel to the minor sheet (Fig. 2B, red). The connecting regions fold into helices (α1–7) or loops, and the C-terminal insertion forms a helix (α8) extending from βVIII. At the open end of the structure, the C-terminal α8 and N-terminal insertion α4 (linking β3 and β4) align nearly parallel to the β-sheets, creating a gate-like structure (Fig. 2B, yellow). Meanwhile, the loop connecting the N-terminal insertion β4 and β5 forms a lid-like configuration (referred to as the lid loop, residues G95 to K114, Fig. 2B, red). Together, these structural features form a flexible, closable architecture and differs from TauD-like decarboxylases, which typically have a major central insertion with a more open, accessible active site[20,21].

At the active site, one monomer displays electron density for Mn²⁺ only (ligand unbound state), while the other shows clear electron densities for Mn²⁺, αKG, and the substrate (ligand bound state, Fig. 2A, **S1A**, and **S1B**). Structural comparisons revealed significant rearrangements between the two monomers (Fig. 2E). In the ligand-unbound monomer, the lid loop is mostly disordered, as evidenced by the lack of defined electron density for residues 97–111 (Fig. S1B), suggesting a dynamic, unstructured state. In contrast, in the ligand-bound monomer, the lid loop is well defined, with clear electron density observed (Fig. S1A). Simultaneously, the gate helices, α4 (residues 76–86) and α8 (residues 307–322), shift inward toward the active site, with residues 317–323, which are also disordered in the unbound state, adopting an ordered conformation upon ligands binding. Key side chains, including R90 and Y214, move outward to avoid steric clashes with the substrate, while Y85 and F130 shift inward. These structural rearrangements drive the transition of active site between an open and a tightly closed state, as revealed by the surface view of the monomers, where the ligands are deeply buried in the ligand-bound monomer (Fig. 2D) and would be exposed when overlaid with the unbound structure (Fig. 2C). In addition, a crystal structure with higher resolution at 2.1 Å, which shows no density of substrate or αKG binding but reveals Mn²⁺ binding in both monomers, confirms the appearance of the unbound conformation (Fig. S1D, PDB ID: 9IG4, Supplementary Data 2). These findings demonstrate that the binding of αKG and the substrate triggers conformational changes, facilitating the tight closure of the active sites.

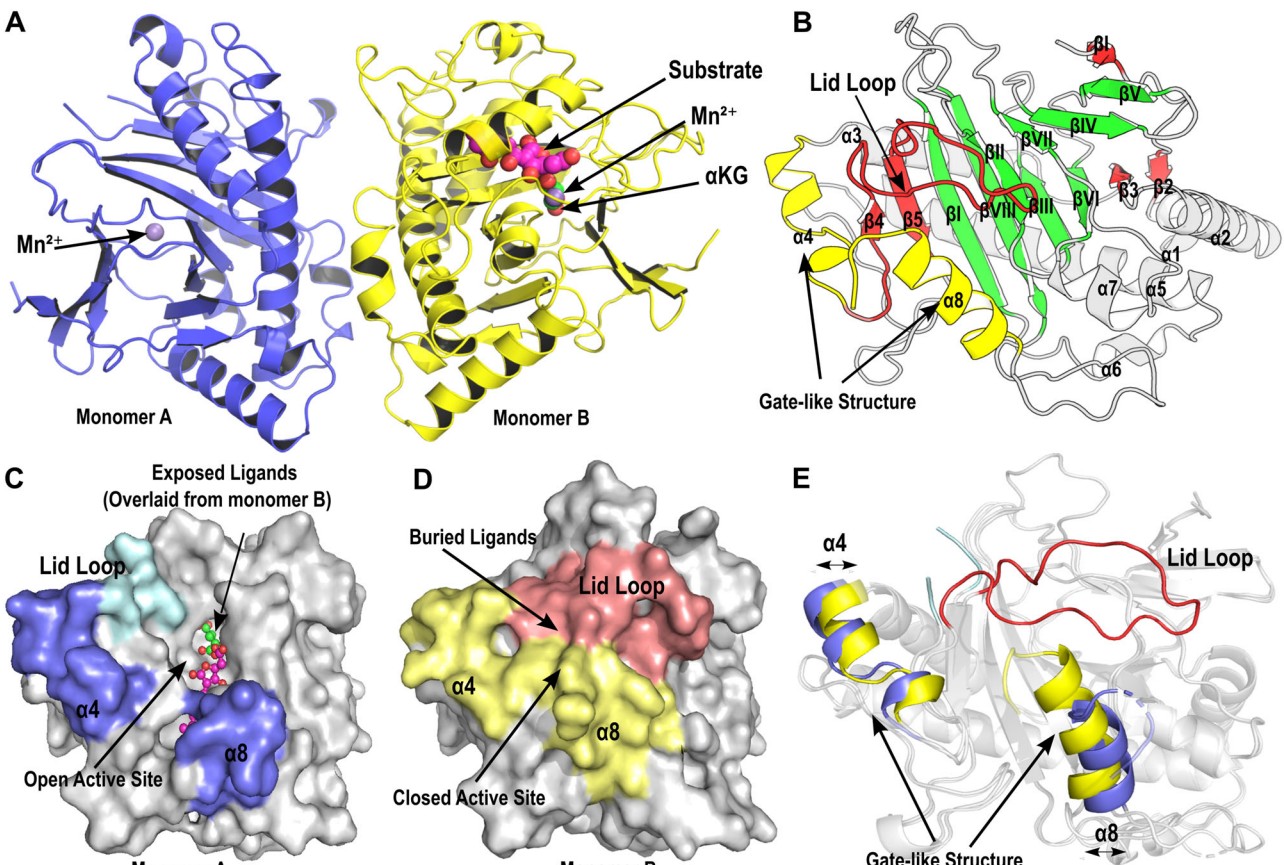

**Fig. 2 | Crystal structure of TraH: monomer comparison and structural features.**
**A** Overall view of TraH crystal structure. $Mn^{2+}$ is shown as a light blue sphere binding to both monomers, while αKG and the substrate are represented as cyan and green ball-and-stick models, respectively, binding exclusively to monomer B, which is referred to as the ligand-bound monomer. **B** Monomer B with labeled β-sheets and α-helices. The β-sheets forming the DSBH core are highlighted in green, while the β-sheets and lid loop from the extra N-terminal insertion are colored tv-red. The gate-like helices α4 (N-terminal insertion) and α8 (C-terminal insertion) are colored

yellow.Surface views of the ligand-unbound monomer A (panel **C**) and the ligand-bound monomer B (panel **D**). The gate-like helices α4 and α8 are highlighted in slate and pale yellow, respectively, in monomer A and B. The lid loop is colored pale cyan in monomer A and salmon in monomer B. In panel **C**, ligands from monomer B are overlaid and displayed in the active site to highlight the location of the active site in the ligand-unbound monomer A. **E** Structural comparison of the two monomers of TraH in cartoon representation. The gate-like helices α4 and α8, along with the lid loop, are colored as in panels (**C** and **D**).

## The lid loop shapes a hydrogen bond network for the recognition and decarboxylation of crustosic acid

Our previous study demonstrated that the substrate, crustosic acid, exists in a dynamic equilibrium between its closed (**1a**) and open (**1b**) states (Fig. 3A)[10]. Crystallographic modeling revealed that the electron density is more consistent with the open conformation (Fig. 3B). As shown in Figs. 3C, D and 4D, the substrate is sandwiched between the two layers of the DSBH core, adjacent to both the metal ion, coordinated by the conserved 2-His-1-carboxylate triad, and αKG, stabilized by the residues R280 and Y195. Residues such as F286 and Y214 mediate π–π stacking interactions with the tetronate core, effectively acting like "clamps" that stabilize substrate binding and positioning, analogous to the role of residues observed in homologous enzymes[16], while K191, H288, Y214, and G215 stabilize the substrate via hydrogen bonds, either directly or through water molecules. The lid loop covers the substrate from the direction of the carboxyl group, sealing the complex and enhancing the stability of the enzyme-substrate interaction. A key feature of this interaction network is Q99, which forms direct hydrogen bonds with the substrate's O1 (3.0 Å) and O10 (3.4 Å). Moreover, Q99 coordinates extensive hydrogen-bond interactions between residues from both the core and the lid loop, including K191 (2.8 Å), Y214 (3.0 Å), D112 (via K114, 3.0 Å), and K114 (2.7 Å). Additionally, the lid loop residue N102 engages the carboxyl group of the substrate through a hydrogen bond with O9 (3.1 Å). A water molecule (Figs. 3B and 3D) is also integrated

into this network, mediating interactions between O1 (2.8 Å) and O8 (3.0 Å) of the substrate with K191 (2.7 Å) and Q99 (3.4 Å), respectively.

To assess the functional contribution of the identified hydrogen-bond network (Q99, D112, K114, K191, Y214) to decarboxylation, we conducted a series of site-directed mutagenesis experiments. The mutational strategy was designed to systematically probe the steric, electrostatic, and hydrogen-bonding requirements at each position (see Table S2 for a summary of the design considerations). All enzymatic activities were measured using the closed-form substrate **1a** and its corresponding product, which predominates under the assay conditions (Fig. S2). Focusing on residue Q99, pronounced differences in catalytic activity were observed among the tested variants. Substitutions such as Q99E, Q99S, Q99K, Q99L and Q99R either completely abolished activity or retained only trace activity. In contrast, the Q99H variant maintained moderate activity. Collectively, variants that preserved side-chain features most similar to glutamine—particularly hydrogen-bonding capacity and appropriate side-chain volume—exhibited higher residual activity, consistent with a dual steric and hydrogen-bonding role for Q99 within the network. Mutations at other residues involved in the hydrogen-bond network, including D112, K114, K191, and Y214, also led to substantial reductions in decarboxylation activity (Fig. 3E). The K114M and K191M mutants showed no detectable activity, whereas variants such as D112L, K114R, K191R, Y214F, and Y214H exhibited weak residual activity, highlighting the importance of both electrostatic interactions and properly organized hydrogen-bonding interactions for efficient catalysis.

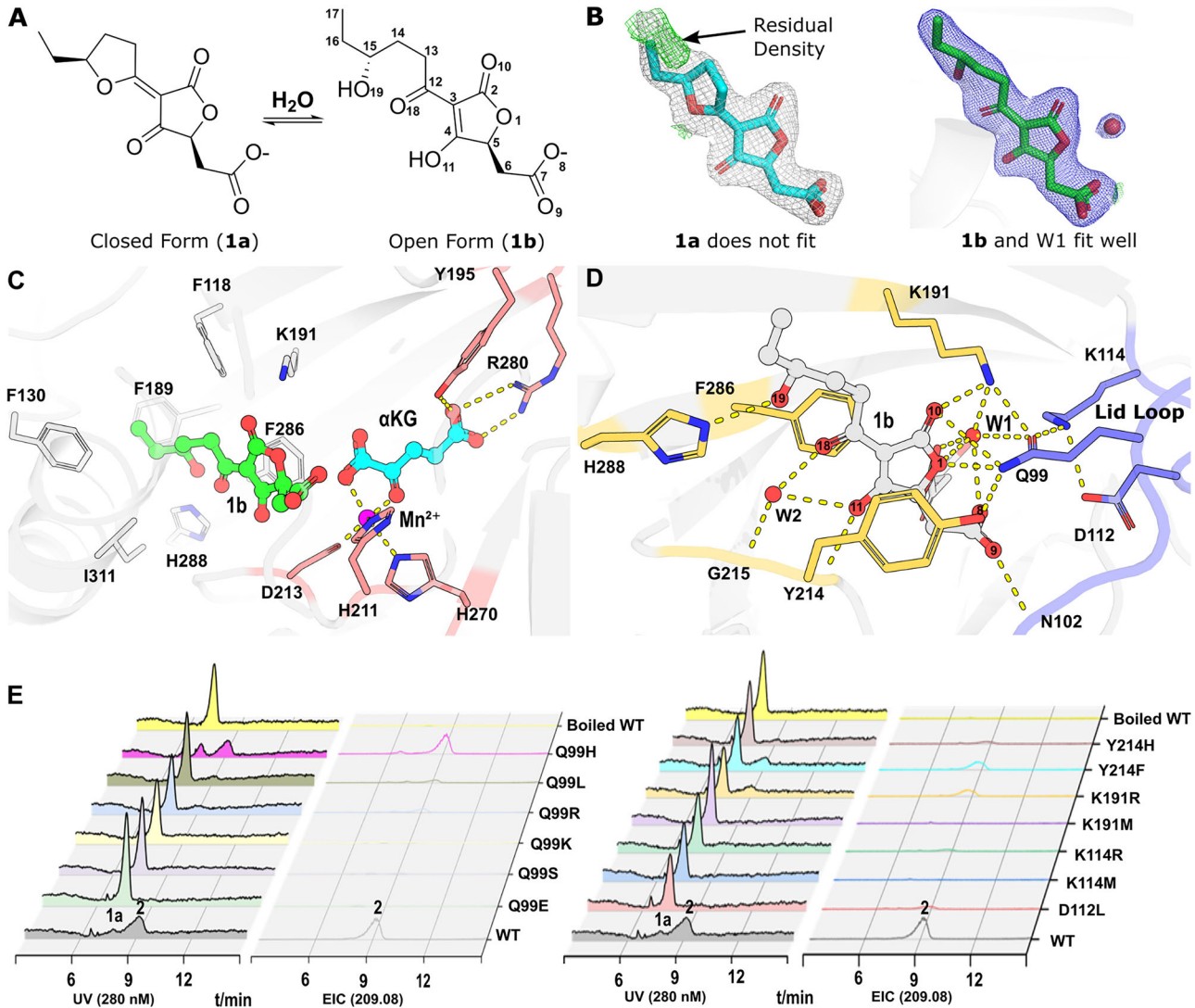

**Fig. 3 | Active site of TraH and mutational impact on crustosic acid decarboxylation. A** Equilibrium between closed (**1a**) and open (**1b**) forms of crustosic acid. **B** Modeling crustosic acid into the active site of TraH. The post-modeling 2mFo-DFc map at 1.5σ (carve = 1.5, blue) shows well-defined electron density for open form **1b** and the associated water molecule, with no significant residual density in the mFo-DFc map (2.5σ, carve = 1.5). In contrast, the mFo-DFc map (green, 2.5σ, carve = 1.5) reveals substantial residual electron density for closed form **1a**, despite the reasonable density observed in the 2mFo-DFc map (gray, 1.5σ, carve = 1.5), indicating a poorer model fit. **C** Overall view of TraH active site. Mn²⁺ is shown in magenta and αKG in cyan, with metal-coordinating residues and αKG-stabilizing residues in

pink. The substrate is shown in green. **D** Detailed view of substrate interactions highlighting the lid loop-mediated hydrogen bond network. The substrate is shown as a gray ball-and-stick model, key residues from the DSBH core are in light yellow, while the lid loop and residues from this region are in light blue. **E** Decarboxylation activity of key residue mutants, including Q99, D112, K114, K191, and Y214. Reaction samples were analyzed by HPLC-MS, with the corresponding UV absorption at 280 nm and EIC chromatograms. Both open- and closed forms were monitored (Fig. S2), but only the closed form was predominantly detected ([M + H]⁺ at m/z 209.08). Source data are provided in Supplementary Data 4.

Importantly, the specificity of this network was further supported by control mutations targeting nearby loop residues that do not participate in direct hydrogen-bonding interactions with the substrate (E98, Q100, and S208). None of these mutations resulted in significant loss of decarboxylation activity (Fig. S3A), indicating that the observed severe losses in catalytic activity arise from disruption of the specific hydrogen-bond network rather than from global structural perturbations.

### Hydrogen-bond network is not necessary for desaturation of crustosic acid methyl ester catalyzed by TraH

In addition to catalyzing the decarboxylation of crustosic acid (**1a**), TraH also facilitates the desaturation of its methyl ester (**3a**) in vitro (Fig. 4A)[10]. To better understand the catalytic mechanism of TraH, we co-crystallized TraH with Mn²⁺, αKG, and **3a**, which was observed in its open state (**3b**, Fig. 4A) in the complex structure (TraH•Mn²⁺•αKG•**3b**, PDB ID: 9IG3, Supplementary

Data 3). Comparison of the structures bound to **1b** and **3b** revealed a similar overall fold, but notable differences in the conformation of the lid loop region (Fig. 4B, D). In the **3b**-bound form, residues 100 – 112 exhibit disorder, resembling their unbound state, while the side chains of Q99 and K114 are slightly displaced (Fig. 4B, D). As a result, their involvement in the hydrogen bond network observed in the **1b**-bound structure is disrupted in the **3b**-bound state. As expected, mutations in the lid loop residues involved in hydrogen bond network, such as Q99, D112, and K114, which significantly reduce decarboxylation activity, have limited impact on desaturation of **3a** (Fig. 4C). For example, Q99E, Q99S, Q99K, and K114M exhibit undetectable decarboxylation activity for **1a**, while their desaturation activity for **3a** remains prominent. Meanwhile, D112L and K114R retain weak decarboxylation activity, but their desaturation activity for **3a** is largely unaffected. These results suggest that the lid loop is not essential in the desaturation of crustosic acid methyl ester.

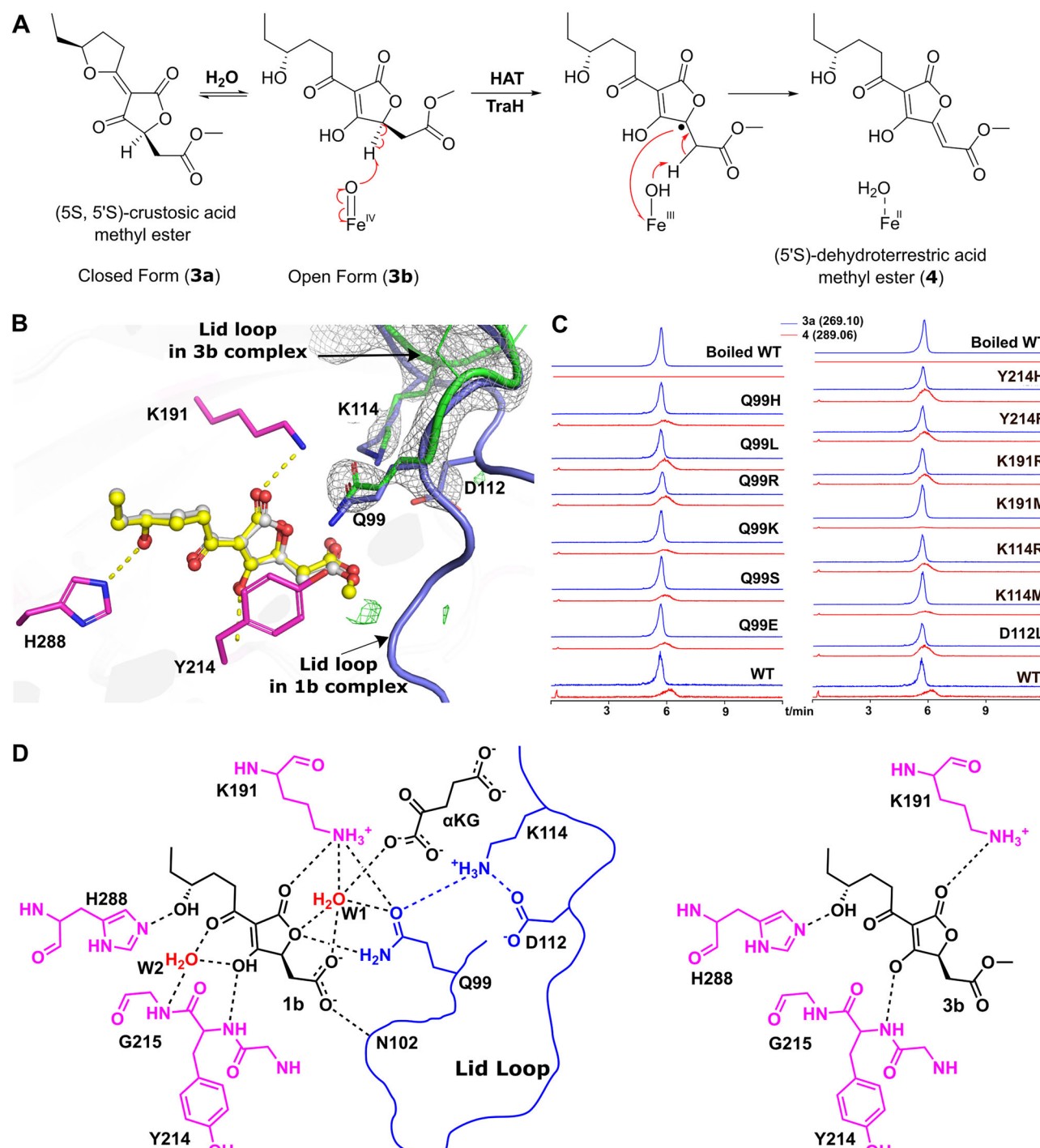

**Fig. 4 | TraH-catalyzed desaturation and lid loop involvement in comparison to decarboxylation. A** Desaturation of (5S,5'S)-crustosic acid methyl ester catalyzed by TraH and its proposed mechanism. **B** Detailed view of the interaction between TraH and **3b**. **3b** is shown in a yellow ball-and-stick representation, while key residues in the DSBH core involved in **3b** binding are highlighted in magenta. The partially resolved lid loop is colored green, with the corresponding 2mFo-DFc map (gray, 1.2σ, carve = 1.5) and residual mFo-DFc map (green, 2.5σ, carve = 1.5) in the loop area. The structure of the **1b** complex was superimposed for comparison, with **1b** shown in gray and its associated lid loop in blue. **C** Desaturation activity of key

residue mutants within the hydrogen bond network. Reaction samples were analyzed by HPLC-MS, and the corresponding EIC chromatograms are shown for substrate **3a** (m/z 269.10, [M + H]⁺, blue) and product **4** (m/z 289.06, [M+Na]⁺, red), Source data are provided in Supplementary Data 4. **D** Schematic representation of hydrogen bond interactions between TraH and its substrates. The interactions for **1b** and **3b** are shown in the left and right panels, respectively. Ligands are colored in black, residues in the DSBH core are highlighted in magenta, and the lid loop along with its residues are shown in blue.

Despite these differences, the binding of **3b** closely resembles that of **1b** in several respects. Both the α4 and α8 helices maintain similar orientations, and side chains such as Y85, R90, F130, and Y214 undergo similar conformational changes upon binding either substrate. Notably, the hydrogen

bond between K191 and the backbone nitrogen of Y214 remains unchanged (Fig. 4B, D). Although the role of the hydrogen bond between Y214 and **3b** cannot be directly verified, the substantial reduction in activity observed with the K191M mutant underscores the critical role of K191 for binding

and orientating substrate **3b**, comparable with its involvement in the decarboxylation reaction (Figs. 3E, 4C). In contrast, the K191R, Y214H, and Y214F mutants show little impact on the desaturation of **3a** but exhibit a noticeable reduction in the decarboxylation of **1a** (Figs. 3E, 4C). These results suggest that while these mutations preserve interactions necessary for desaturation, their altered side-chain structures likely disrupt the hydrogen bond network essential for decarboxylation. These findings highlight the mechanistic differences between decarboxylation of crustosic acid and the desaturation of the related methyl ester by TraH. While decarboxylation relies on a well-ordered hydrogen bond network, desaturation appears to proceed independently of this network, demonstrating the enzyme's flexibility in accommodating and catalyzing reactions of chemically diverse substrates.

## Hydrogen bond network mediates proton transfer during decarboxylation process

To investigate the role of the lid loop in the decarboxylation process, we combined molecular dynamics (MD) and quantum mechanical/molecular mechanical (QM/MM) metadynamics simulations to explore the catalytic mechanism, with a focus on the process after the formation of the $Fe(IV) = O$ intermediate, which is a well-established species in reactions catalyzed by $Fe/\alpha KG$-dependent enzymes[22] (Fig. S4A). Previous studies have suggested that basic residues often play a crucial role in the decarboxylation process of iron-dependent decarboxylases[5,20,23]. For example, in SocE, a protonated R310 has been proposed to act as a proton donor during catalysis. In TraH, basic residues such as K191 and K114 are present in the active site but do not directly interact with the carboxyl group of the substrate. Furthermore, they are located far away from the iron center. These observations led us to hypothesize that TraH might facilitate decarboxylation via a water-mediated mechanism, similar to what is observed in cytochrome P450 enzyme OleT[23]. Indeed, post-equilibration MD simulations of the TraH•Fe(IV) = O•succinate•**1b** complex revealed a stable and persistent water network (W1 and W2) adjacent to K191. The position and orientation of this network were highly similar to that observed in the crystal structure (Figs. S4B, S4C, S5D, S5E), confirming its relevance in the solution dynamics of the reactive complex. Based on this structural evidence for a pre-organized proton-transfer pathway, we extended the QM region beyond the immediate iron coordination sphere (H211, D213, H270), succinate, and the substrate, to explicitly include the side chain of K191 and water molecules W1 and W2, in order to investigate their potential role in the decarboxylation process.

The decarboxylation reaction was proposed to proceed in three steps (Fig. 5A): (1) Hydrogen atom abstraction (HAT) by $Fe(IV) = O$ from the substrate, reducing $Fe(IV)$ to $Fe(III)$-OH.; (2) Single electron transfer (SET) from the substrate radical to $Fe(III)$-OH, converting it to $Fe(II)$-OH and triggering substrate decarboxylation with $CO_2$ release; and (3) Proton transfer and $Fe(II)$ restoration, where $Fe(II)$-OH accepts a proton, restoring to its $Fe^{2+}$ state and completing the catalytic cycle. Following this step, K191 is re-protonated, likely via proton transfer from nearby solvent water molecules (W1/W2) observed in the MD simulations, regenerating the initial state for the next catalytic cycle. The free energy profile and representative snapshots along the reaction coordinate (Figs. 5B and S7) illustrate a feasible reaction pathway, showing stepwise progression from the reactant complex (RC) through transition states (TS1, TS2, and TS3) to intermediates (INT1 and INT2), and finally to the product (P), with calculated energy barriers of 14.1, 8.3, and 4.8 kcal/mol for each step. K191 and the water molecules play critical roles in steps 2 and 3. In step 2, water molecule W2 forms a hydrogen bond with Fe (II)-OH, stabilizing the transition state TS2 (Fig. S7B). In step 3, the proton from water molecule W2 migrates to Fe(II)-OH, forming water molecule (W3). Concurrently, protonated K191, acting as a catalytic acid, donates a proton to W1, facilitating proton transfer through the water network and completing the catalytic cycle. During this step, the Fe–O bond elongates to 2.48 ± 0.34 Å (Table S3), suggesting the dissociation of the oxygen atom from the iron center, which facilitates the recovery of the iron center to its ground state, ready to proceed with the next

catalytic cycle. These findings underscore the pivotal role of water-mediated proton transfer in facilitating the catalytic cycle.

A detailed analysis of the pre-decarboxylation complex and key reaction stages highlights the role of the lid loop in the decarboxylation process. During MD simulations of the pre-decarboxylation complex, the persistent presence of the water chain, which provides a critical starting point for the subsequent decarboxylation process, is observed to be stabilized by the lid loop through hydrogen bonding interactions (Fig. 6A), similar to those observed in the crystal structure (Figs. 3D, 4D). Specifically, lid loop residue Q99 is observed to interact with the substrate carboxyl group and form a hydrogen bond with K191 and D112, as illustrated in the representative snapshot. Additionally, lid loop residue N102 forms a hydrogen bond with the substrate carboxyl group, which, together with K191, plays a crucial role in positioning the substrate, and water molecules W1 and W2 through hydrogen bonds. In the QM/MM simulations of the decarboxylation process, key stages, such as the transition state TS2 and proton migration to W3, are characterized by the involvement of the water chain, while K191 moves within hydrogen bonding range of the lid loop residue K114, potentially facilitating proton transfer (Fig. 6B). These results are in agreement with mutational studies showing that disrupting this network leads to the loss of decarboxylation activity for crustosic acid (Fig. 3E). Taken together, these findings underscore the indispensable role of the lid loop in maintaining an optimal water-mediated proton transfer network, which is crucial for efficient decarboxylation catalysis. The lack of a similar requirement for the lid loop in the desaturation of crustosic acid methyl ester suggests that the dehydrogenation may proceed via a different mechanism, potentially involving hydrogen abstraction from the adjacent carbon of the substrate radical itself, as reported in the other $\alpha KG$ dependent dehydrogenases[20,22] (Fig. 4A).

## Lid loop conservation and divergence potentially contribute to substrate binding and specificity

Our analysis revealed a critical role of the TraH lid loop in substrate binding and decarboxylation, suggesting that this region is also likely important for substrate recognition and specificity in related $\alpha KG$-dependent dioxygenases. Structural comparisons of TraH with homologs identified through DALI analysis (Table S5), including TropC (PDB ID: 6XJJ), T7H (PDB ID: 5C3R), and EFE (PDB ID: 5V2Z), reveal a pronounced lid loop (Fig. S8). This loop adopts a closed conformation in TropC (even without $\alpha KG$ and substrate bound) or closes upon ligands binding in EFE, where it participates in catalysis or substrate binding[18,24], but remains open and functionally disengaged in T7H[16]. In contrast, the structurally homologous enzyme IPNS (PDB ID: 1BK0) possesses a much shorter lid loop that remains open and does not contribute to catalysis or substrate interaction[15] (Fig. S7B). To explore the evolution and conservation of the lid loop in more detail, we next inferred a phylogenetic tree of TraH and related fungal $\alpha KG$-dependent dioxygenases. This analysis allowed us to examine the conservation of critical residues involved in the hydrogen network forming via the loop region.

Using homologous sequences from green algae as an outgroup, our rooted phylogeny displays two major paralogous clades (clade A and clade B) that can mostly be assigned to Pezizomycotina. Clade B harbors enzymes of unknown function, and clade A contains TraH and other related enzymes with well-established catalytic roles[10,11,17,25–28] (Fig. 7A). Within both clades, we noticed several additional putative gene duplications and losses. Frequent lineage specific duplications and losses are also common in other fungal secondary metabolic enzymes[29]. Focusing on clade A, the phylogeny groups those enzymes to which catalytic functions have been assigned further into various paralogous clades (A1 to A4): ClaD/CitB (hydroxylation) are placed within clade A2, TropC/Asl3/EupD (ring-expansion) fall into clade A3 and TraH/ThnC (decarboxylases) belong to clade A4. We next investigated the sequence conservation of the key residues that participate in substrate binding and catalysis across these clades: Q99, A101, D112, and K114 from the lid loop region, as well as K191 and Y214 from the DSBH core. To compare the conservation of these residues across clades, we computed clade-specific sequence logos of the corresponding motifs. These

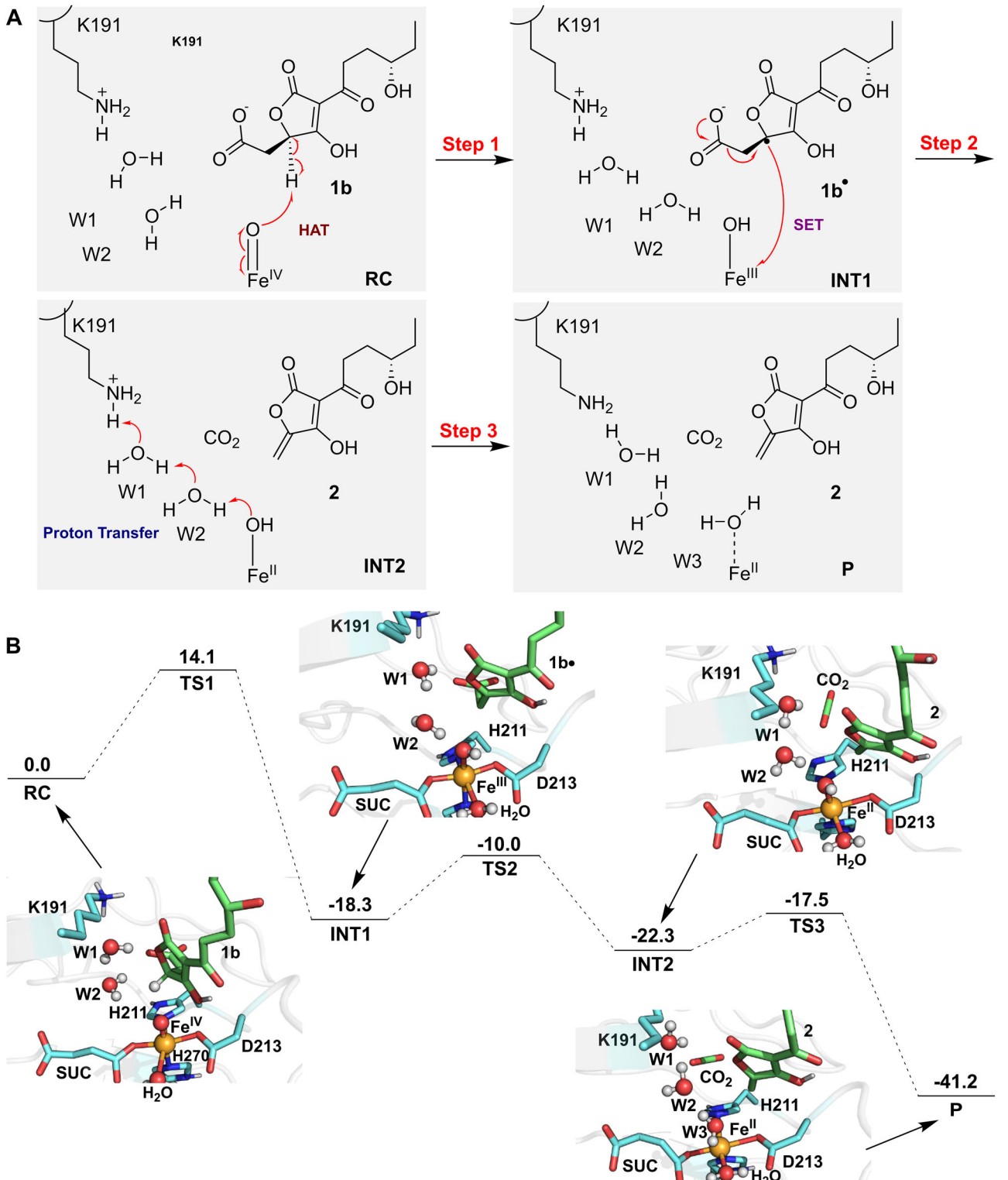

**Fig. 5 | Mechanistic insights into TraH-catalyzed decarboxylation via QM/MM metadynamics simulations. A** Predicted catalytic mechanism of TraH in the decarboxylation of crustosic acid. **B** Free energy profile (kcal mol⁻¹) and representative snapshots along the reaction coordinate. The carbon atoms of **1b**, **1b•**, or **2** are colored green, key residues in cyan, and the iron center in orange. Relevant hydrogen bonds are depicted as black dashed lines, while covalent bonds undergoing formation or cleavage are highlighted in red. Detailed structures of transition states TS1, TS2, and TS3 are provided in Fig. S7. Average key active-site distances and representative spin density values along the reaction pathway are summarized in Tables S3 and S4, respectively.

reveal that while the DSBH sites K191 (either K or R) and Y214 (either Y or F) exhibit conservation of chemical properties, the lid loop residues, D112 and K114, are highly conserved across all paralogous clades. Q99 can in the TraH/ThnC (A4) clade be substituted to either glutamine, serine, or

threonine but is fully conserved across the ClaD/CitB (A2) and the TropC/Asl3/EupD (A3) clades. Interestingly, position 101 is much more divergent both across and also within clades and can exhibit alanine, leucine, histidine, arginine, or proline.

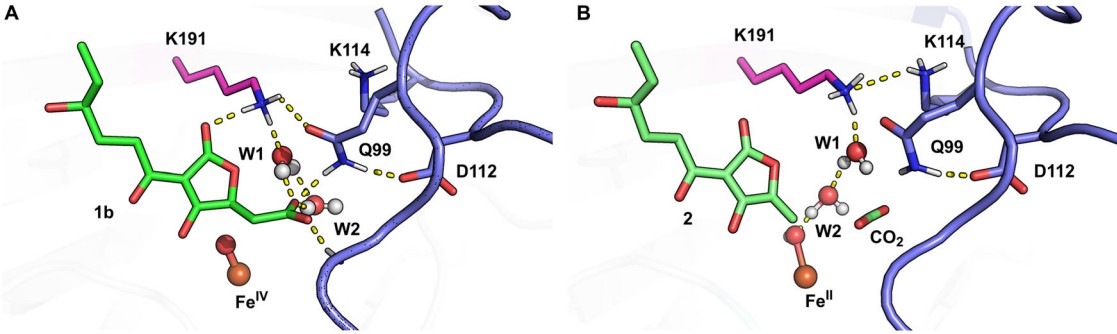

**Fig. 6 | Involvement of the lid loop in pre-decarboxylation complex and key stages during TraH-catalyzed decarboxylation.** Panels (**A**) and (**B**) show the detailed views of the lid loop involvement in the pre-decarboxylation complex (TraH•-Fe(IV) = O•succinate•**1b**, derived from a representative snapshot of MD simulations) and during proton migration to W3, respectively. The lid loop and related key residues are highlighted in blue, while key DSBH residue in magenta. The iron ion is displayed in orange, with **1b** or **2** (and $CO_2$) in green. The hydrogen bonds are represented by yellow dashed lines.

To explore the roles of these residues in more detail, we carried out structure prediction of representative enzymes together with their known substrates. The predicted models suggest that the lid loops in clade A enzymes likely remain in a closed or nearly closed conformation and that the aforementioned residues participate in substrate binding (Fig. 7B–E). In TraH, although A101 does not directly participate in substrate binding, the backbone of N102 forms part of the hydrogen bond network mediated by Q99. This interaction is likely facilitated by the small side chain of A101, bringing the backbone closer to the substrate. These findings suggest that the lid loop regions in these enzymes may have evolved to provide additional stabilization and enable precise substrate positioning, particularly for substrates with polar oxygen atoms. The high conservation, or in some cases chemical properties, for most of the key residues (D112, K114, K/R191, and Y/F214) in TraH indicates that these amino acids are, indeed, most likely critical for related αKG-dependent dioxygenases. In contrast, the plasticity at sites 99 and 101 suggests that these residues might contribute to the substrate specificity that appears to differ across the paralogous clades. Indeed, our mutational analysis showed that mutating Q99 in TraH to a serine mostly abolished crustosic acid decarboxylation (Fig. 3E). Given that S99 is found in almost half of the other A4 clade enzymes, this suggests that position 99 may confer substrate-specificity in TraH and closely related enzymes, potentially in combination with variations at other residues such as A101 (L, Y, H, A/P, or R), K191 (K or R), and Y214 (Y or F).

Further analysis of lid loops in available crystal structures from the IPNS subfamily revealed a widespread presence of lid loops (except in ACCO and HIS1), with considerable variability in composition, length, and conformation (Table S6). In some enzymes, such as DAO, DPS, IPNS, T6ODM, DAOCS, and T7H, lid loops are observed in open or disordered states even with substrates binding, suggesting they do not directly contribute to substrate binding or catalysis[15,16,30,31]. In contrast, other enzymes feature closed or closable lid loops, typically 15–16 residues long, which likely assist in substrate binding. For instance, lid loop residues E116 and K129 in H6H and E84 in EFE form critical hydrogen bonds with the substrate[18,24,32]. Similarly, the 'gate' loop in G2OX3 likely alternates between open and closed conformations to facilitate substrate entry and stabilization[33]. These findings highlight the evolutionary flexibility of lid loops, which not only stabilize substrate binding but also enable functional diversification to accommodate distinct substrate requirements across the IPNS subfamily. The adaptability of the lid loop underscores its critical role in fine-tuning enzymatic activity, particularly in the context of polar substrate recognition and stabilization.

## Discussion

The αKGDS enzymes are known for their remarkable ability to recognize and oxidize a wide variety of structurally diverse substrates, a versatility enabled by their conserved DSBH core structure and flexible regions that enable adaptation to diverse substrate features[1,14]. TraH, a representative member of this family, catalyzes the oxidative decarboxylation of crustosic acid from *Penicillium crustosum*. Structural analysis reveals that the DSBH core facilitates substrate binding by utilizing its extended β-sheet layers to accommodate the substrate and orient its carboxylate group toward the protein surface, where the N-terminal lid loop provides essential shielding. Within this loop, Q99 participates in a hydrogen-bond network that stabilizes water molecules interacting with the conserved K191 residue in the core. This network enables proton transfer from K191 via the water molecules to the iron-coordinated hydroxyl, thereby facilitating regeneration of the resting state, as supported by mutagenesis and QM/MM metadynamics simulations. In contrast, other αKG-dependent decarboxylases and dioxygenases, such as ACCO, IsnB, and ScoE, typically rely on core-located basic residues (arginine or lysine) to directly stabilize the departing carboxyl group and trigger decarboxylation[5,20,24]. The TraH mechanism therefore represents a distinct decarboxylation strategy within the αKGDS family, where catalysis is orchestrated remotely through loop-mediated proton transfer, rather than executed directly within the active-site pocket. This likely reflects an adaptation for accommodating extended carboxylate-containing substrates.

Given that perturbation of the hydrogen-bond network may also influence substrate binding or reactive positioning, these structural effects must be considered when interpreting its central role in mediating proton transfer during decarboxylation. A quantitative assessment of binding affinity, for example by isothermal titration calorimetry, would be informative but is currently limited by substrate availability. Nevertheless, structural analysis indicates that the corresponding methyl ester substrate, which lacks the carboxylate group required to engage this hydrogen-bond network, adopts a broadly similar overall binding pose within the active site. Consistent with this observation, most examined mutants show only minimal effects on the desaturation activity toward the methyl ester substrate. Therefore, while we cannot fully exclude some influence on substrate binding, the combined evidence strongly suggests that the hydrogen-bond network functions primarily to coordinate proton transfer by assembling the proton-transfer apparatus in a productive geometry, thereby regulating the catalytic outcome in a substrate-specific manner. At a broader mechanistic level, stepwise descriptions involving the development of carbocation-like character have been proposed for related enzymes such as PlsnB[5], in which such intermediates are likely stabilized by conjugation with an adjacent aromatic ring. While our QM/MM meta-dynamics simulations do not reveal a distinct free-energy minimum corresponding to a long-lived carbocation intermediate, alternative stepwise electronic descriptions cannot be fully excluded and may represent limiting cases of a continuous reaction coordinate in high-valent iron–oxo chemistry[20,22]. Moreover, our evolutionary and structural analyses reveal that the lid loop is widely conserved across the IPNS family but also shows variability at key sites, with distinct sequence patterns among different subclades of TraH-related enzymes.

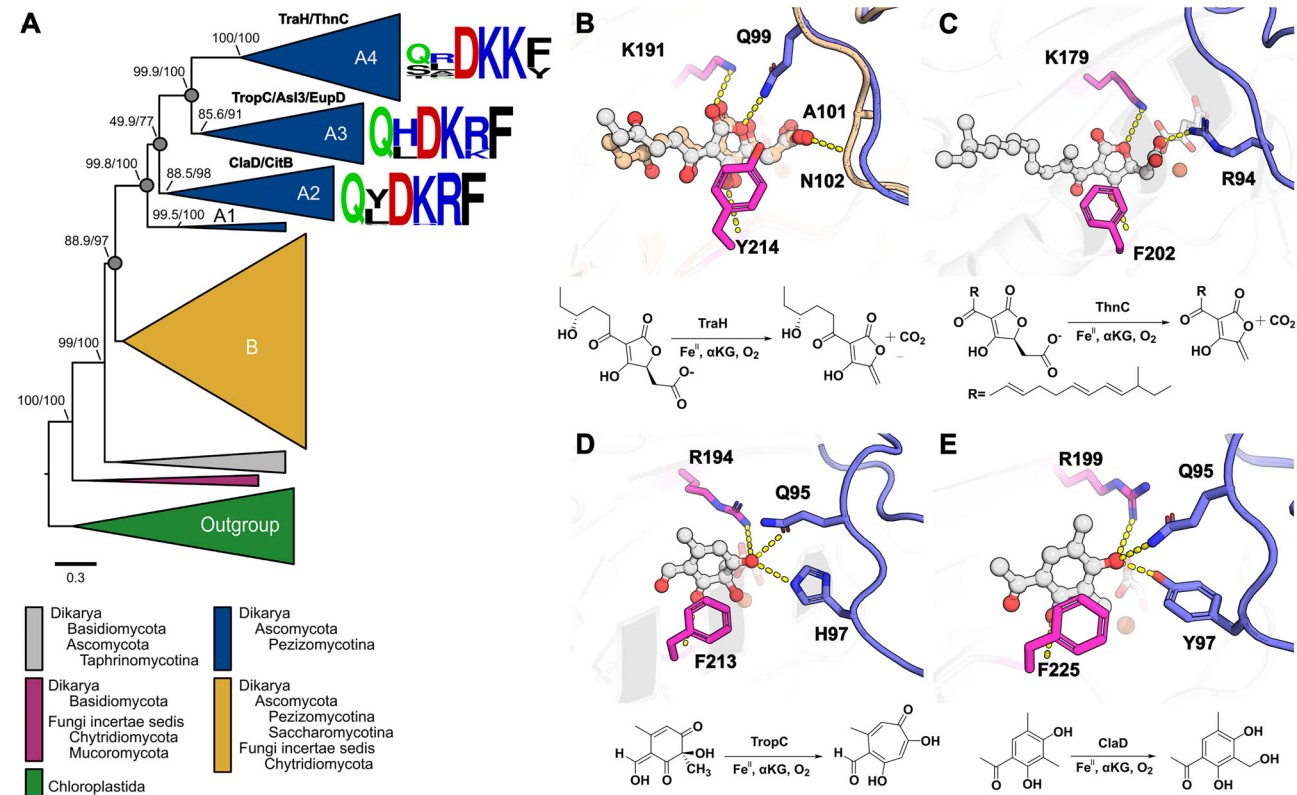

**Fig. 7 | Phylogenetic relationships, catalytic reactions, and lid loop-related structural adaptations in TraH homologs. A** Maximum-likelihood phylogenetic tree of TraH homologs with sequence logos for the subclades and regions of interest. The clades of interest include A2, A3, and A4, which encompass the experimentally characterized enzymes CitB/ClaD, TropC/Asl3/EupC, and TraH/ThnC, respectively. The regions of interest, corresponding to positions Q99, A101, D112, K114, K191, and Y214 in TraH, feature residues involved in the loop-mediated hydrogen bond network within TraH. The phylogeny was rooted using TraH homologs from Chloroplastida (green algae) as an outgroup. The clades are colored to depict either species coverage within clades or to distinguish the paralogous clades A and B. Gray spheres indicate assigned duplication nodes. Nodes are labeled with SH-like

approximate likelihood ratio test (SH-aLRT) and ultrafast bootstrap support values. Scale bar: expected substitutions per site. **B–E** Reactions catalyzed by representative experimentally characterized enzymes and their predicted substrate binding modes. The predicted substrate binding models for TraH, ThnC, TropC, and ClaD, derived from Chai Discovery, are shown in panels (**B**, **C**, **D**, and **E**), respectively, where the TraH substrate is bound in a manner similar to the crystal structure. The docking structures are shown as gray cartoons with the lid loop highlighted in blue, while the crystal structure is shown in light yellow. Residues involved in hydrogen bonding with the substrate are shown as stick models, with loop residues in blue and residues from the DSBH core in magenta. The substrates are displayed as ball-and-stick models. Hydrogen bonds are indicated by yellow dashed lines.

This coexistence of conservation and variability suggests that the lid loop has evolved to accommodate diverse substrate features while fine-tuning catalytic outcomes. Notably, residues proximal to the active site, including positions corresponding to Q99 and A101, exhibit pronounced subclade-specific variations, implying potential roles in modulating substrate positioning and reactivity. Consistent with this idea, studies of the related enzyme TropC have shown that mutation of the corresponding residue (H97A) leads to altered product profiles[17], highlighting a functional role for this region. Similar loop-mediated regulation has been observed in the clavaminate synthase–like (CSL) subfamily, where substrate-capping loops contribute to substrate binding, recognition, and stereoselectivity in enzymes such as AAD, AsnO, VioC, and KDO[34,35], which adopt a TauD-like structural fold. However, whereas these TauD-family loops typically originate from central insertions[4,14], the IPNS lid loop is positioned at the N-terminus (Fig. S9), representing a distinct structural solution for substrate engagement. Thus, while loop-mediated control of substrate access and positioning appears to be a recurring functional theme among αKG-dependent enzymes, the structural implementations of this control can differ substantially. Collectively, our work on TraH uncovers a previously unrecognized decarboxylation strategy mediated by an N-terminal lid loop, illustrating how a conserved enzymatic scaffold can evolve divergent structural solutions to achieve precise substrate control and expanding our understanding of functional diversity within this enzyme superfamily.

## Methods

### Protein expression and purification

Mutations in the *traH* gene were introduced using the QuikChange site-directed mutagenesis method with specifically designed primers for PCR amplification of the plasmid containing the *traH* gene (Table S7). Protein expression was induced by lactose auto-induction in LB medium supplemented with 50 mg/ml kanamycin, with shaking at 30 °C for 16–20 h. Cells were harvested by centrifugation and lysed using an LM10 microfluidizer (Microfluidics) at 15,000 psi with lysis buffer (20 mM HEPES, 150 mM NaCl, 40 mM imidazole, pH 8.0). Recombinant protein was purified using a HisTrap HP 1 mL column (Cytiva) with an elution buffer (20 mM HEPES, 150 mM NaCl, 200 mM imidazole, pH 8.0). The protein was further purified by gel filtration chromatography on a HiLoad 26/600 Superdex 200 pg column (Cytiva) equilibrated with storage buffer (20 mM HEPES, 200 mM NaCl, pH 7.5), following removal of chelated metal ions by EDTA. The TraH mutant proteins were expressed and purified using the same procedure.

### Crystallization, structure determination, and analysis

WT TraH protein was concentrated to 40 mg/ml (determined by absorption at 280 nm) and supplemented with $MnCl_2$, αKG, and substrate in a molar ratio of 1:2:2:2 (0.6:1.2:1.2:1.2 mM) for crystallization. Crystallization was performed at 20°C using a sitting-drop vapor diffusion method. Each drop contained 0.25 μl of protein solution and 0.25 μl of reservoir solution,

equilibrated against 35 µl of reservoir solution. Crystals of TraH•Mn²⁺•αKG•**1b** and TraH•Mn²⁺•αKG•**3b** formed within three days with a solution of 0.1 M CHES, pH 9.5, 30% (w/v) PEG 3000. In contrast, TraH•Mn²⁺ crystals were obtained in 0.1 M sodium cacodylate, pH 6.5, 5% (w/v) PEG 8000, 40% (v/v) MPD. Diffraction data were collected at the P13 beamline of the Hamburg Synchrotron Radiation Facility (HASYLAB), Hamburg, Germany. The data were indexed, integrated, and scaled using XDS[36]. The initial phase was determined by molecular replacement with Phaser MR from the CCP4 software suite, using structural predictions from AlphaFold2 as the search model[37,38]. The phase was optimized through interactive model correction with COOT and refinement with PHENIX until the $R_{work}$ and $R_{free}$ values converged[39,40]. The final model was validated using MolProbity. Data processing and refinement statistics are provided in Supplemental Table S1. All structural figures were generated with PyMOL. Structural modeling of substrate-bound TraH homologs was performed using the Chai-I webserver[41].

### Enzyme activity assay with HPLC–MS detection

TraH WT and mutant proteins were prepared in storage buffer. Ascorbic acid, Fe[(NH₄)₂(SO₄)₂] (Fe$^{II}$), αKG, and dithiothreitol (DTT) were dissolved in H₂O. Each enzyme assay (50 µL) contained phosphate buffer (20 mM, pH 7.4), ascorbic acid (1 mM), substrate (0.5 mM), DTT (1 mM), Fe[(NH₄)₂(SO₄)₂] (1 mM), αKG (1 mM), glycerol (0.5 − 5%), DMSO (5%), and protein (5.4 µM). The enzyme assays were carried out at 37 °C for 30 min and terminated with one volume of acetonitrile. The reaction mixtures were centrifuged at 17,000 x *g* for 30 min before further analysis on LC-MS.

LC-MS analysis was performed on an Agilent 1260 HPLC system equipped with a Bruker microTOF-QIII mass spectrometer by using VDSpher PUR100 C18-M-SE column (150 × 2.0 mm, 3 µm, VDS optilab Chromatographie Technik GmbH) for separation. Water (A) and acetonitrile (B), both with 0.1% (v/v) formic acid, were used as solvents at flow rate of 0.3 mL/min. The substances were eluted with a linear gradient from 5 - 100% B in 15 min, then washed with 100% (v/v) solvent B for 5 min and equilibrated with 5% (v/v) solvent B for 5 min. UV absorption at 280 nm were illustrated in this study. Electrospray positive ionization mode was selected for determination of the exact masses. The capillary voltage was set to 4.5 kV and a collision energy of 8.0 eV. 5 mM sodium formate was used in each run for mass calibration. The masses were scanned in the range of m/z 100.000–1500.000. Data were evaluated with the Compass DataAnalysis 4.2 software (Bruker Daltonik, Bremen, Germany).

### Molecular dynamics (MD) simulations

All MD simulations were performed using the GPU version of Amber 22 package[42], with the initial TraH•Fe(IV) = O•succinate•**1b** complex constructed based on the crystal structure. The Fe(IV) = O, Fe(III)-OH, and Fe(II)-OH species in TraH were parameterized using the *MCPB.py* tool[43] from AmberTools22. The protein residues were assigned with the Amber ff14SB force field[44], while the substrates were treated with the general AMBER GAFF force field[45]. Partial atomic charges were obtained using the RESP method at the B3LYP/6-31 G* level of theory. The protonation states of titratable residues (His, Glu, Asp) were determined based on $pK_a$ values from PROPKA at pH 7.0[46], complemented by visual inspection of local hydrogen-bond networks. Histidine residues H174, H211, and H270 were protonated at the δ position, whereas H46, H54, H105, H106, H147, H156, H198, H233, and H288 were protonated at the ε position to maintain physiologically relevant interactions. The system was solvated in a rectangular TIP3P water box[47] with a 16 Å buffer around the protein and neutralized with Na⁺ ions.

Following initial energy minimization (5000 steps of steepest descent, followed by 5000 steps of conjugate gradient), the system was gradually heated from 0 K to 300 K over 50 ps in the NVT ensemble, using the Langevin thermostat (collision frequency of 2.0 ps⁻¹) with a weak restraint of 25 kcal mol⁻¹ Å⁻² on the protein atoms. Equilibration continued for 1 ns under NPT conditions using the Berendsen barostat (1 atm). Afterward, all

restraints were removed, and the system was further equilibrated for 2 ns under NPT conditions. A 100 ns production MD simulation was conducted under NPT conditions, using a 2 fs timestep with SHAKE constraints[48] applied to covalent bonds involving hydrogen atoms. Long-range electrostatics were treated with the Particle Mesh Ewald (PME) method[49]. Trajectory analyses were performed using standard tools in Amber and VMD[50]. To ensure statistical robustness, three independent replica simulations were performed for each system. The protein backbone RMSD fluctuations during the MD simulations are presented in Figs S5A–C, Supplementary Data 5 and the initial and final configurations for MD have been supplied Supplementary Data 6.

### Quantum mechanical/molecular mechanical (QM/MM) and metadynamics simulations

Quantum mechanical/molecular mechanical (QM/MM) simulations were performed using the CP2K 6.1 package, which employs the dual Gaussian and plane-wave (GPW) basis set and integrates the QUICKSTEP QM module with the FIST MM driver[51]. A representative snapshot extracted from the MD trajectories was used as the initial structure for these simulations. The QM region encompassed the iron center, succinic acid (SUC), key active site residues (H211, H270, D213, K191, Q99), the substrate, and two explicit water molecules (W1 and W2). This region was described at the DFT(B3LYP-D3) level using the Gaussian double-ζ valence polarized DZVP-MOLOPT-SR-GTH basis set with a density cutoff of 360 Ry and Geodecker-Teter-Hutter (GTH) pseudopotentials. The Auxiliary Density Matrix Method (ADMM)[52] was employed to accelerate Hartree-Fock exchange calculations within the B3LYP functional. The remaining atoms were treated at the MM level, using the same force field as in the MD simulations. To couple the QM and MM regions, dangling bonds were capped with hydrogen atoms, and interactions between the QM and MM subsystems were managed using electrostatic embedding with a real-space multigrid technique. The system was equilibrated for 2 ps without constraints using the Born-Oppenheimer molecular dynamics (BOMD) approach, with a 0.5 fs timestep.

To explore the free energy profile of the decarboxylation reaction, QM/MM metadynamics simulations[53] were performed with CP2K 6.1 and PLUMED2 plugin[54]. To characterize the reaction process, three collective variables (CVs) were defined to describe key bond rearrangements (Figure S5). CV1 tracks the transfer of the substrate H5 by measuring the difference between its distances to substrate C5 and the iron-bound oxygen O1. CV2 represents the distance between C6 and C7 of the substrate, reflecting the progression of the decarboxylation step. CV3 describes the proton (H1) transfer from water molecule W2, defined as the difference between its distances to $O_{W2}$ and the iron-bound oxygen O1. Gaussian hills were applied with widths ranging from 0.1 to 0.2 Å, a height of 1.0 kcal/mol, and a deposition interval of 10 fs between consecutive Gaussians. The first re-crossing criterion was used to reconstruct the free energy profile, revealing the key transition states (TS1, TS2, TS3) along the decarboxylation pathway Supplementary Data 5.

### Phylogenetic analysis

TraH homologs from fungi were retrieved from the National Center for Biotechnology Information[55] via protein BLAST search[55,56] using the TraH sequence from *Penicillium crustosum* as a query sequence. The BLAST searches were carried out using the non-redundant protein sequences (nr) database and selected fungal taxa, which were chosen to cover relevant fungal clades. Specifically, denser sampling was carried out within Pezizomycotina but also more distantly related taxa from Basidiomycota, Taphrinomycotina, and unranked fungal species (*Fungi incertae sedis*) were included to better pinpoint duplication nodes. Protein sequences of selected experimentally characterized members of the IPNS subfamily were also included and downloaded from the UniProt database (as of November 19, 2024)[57]. TraH homologs from green algae, which were used as an outgroup to root the TraH phylogeny, were obtained from the EukProt database v3 database[58]. The EukProt database was downloaded, and the BLAST searches

were performed locally using SequenceServer[59]. Sequences were aligned initially aligned via MAFFT[60]. At later stages, the MAFFT-DASH option[61] was used. Sequences were initially clustered using CD-HIT[62] with a similarity threshold of 0.7, trimmed via trimAl[63] and maximum-likelihood (ML) phylogenetic trees were inferred via FastTree 2[64] to guide initial taxon sampling. The resulting tree was visualized and inspected using FigTree[65]. Multiple sequence alignments were inspected via Jalview[66] or AliView[67], which was also used for manual trimming at later stages. Next, IQ-TREE 2[68] was used for inference of ML phylogenies. ModelFinder[69] was used for the selection of the substitution model, which was selected to be LG[70] and a gamma model[71] with 4 rate categories (LG + G4) according to the corrected Akaike information criterion. Statistical branch support values were obtained via the SH-like approximate likelihood ratio test (1000 replicates)[72] and the ultrafast bootstrap approach[73]. The ML phylogeny was rooted using the TraH homologs from green algae as the outgroup. Duplication nodes were assigned based on taxon coverage across clades. Sequence logos were computed using WebLogo[74]. All data used for the preparation of the phylogenetic analysis are provided in Supplementary Data 7.

## Data availability

The atomic coordinates have been deposited in the Protein Data Bank (PDB) under accession code 9IG5 (TraH- Mn-αKG-**1b**); 9IG3. (TraH- Mn-αKG-**3b**) and 9IG4 (TraH- Mn). Other atomic coordinates relevant to this study can be accessed via the PDB under accession codes: 1BK0 (isopenicillin N synthase from *Aspergillus nidulans*); 6XJJ (Structure of non-heme iron enzyme TropC); 5C3R (Crystal structure of the full-length Neurospora crassa T7H), and 5V2Z (Ethylene forming enzyme).

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

## Acknowledgements

We would like to thank the Max-Planck-Society for support (to G.B. and L.Z.). S.-M.L. acknowledges support by the German Research Council (DFG grant ID: INST 160/620-1). X.Z. and R.G. acknowledge the support from China Scholarship Council (201908440573, 202306010090).

## Author contributions

L.Z. and X.Z. contributed to conceptualization, investigation, formal analysis, and writing (original draft and review/editing); G.B. and S.-M.L. for conceptualization, supervision, and writing (review/editing); R.G. and S.-M.L. for biochemical assays; Z.G. and A.L. for MD and QM/MM analysis; M.G. and G.K.A.H. for phylogenetic analysis, and J.F. and A.L. to writing (review/editing). All authors have read and commented to the manuscript.

## Funding

## Competing interests

The authors declare no competing interests.
