## [Transparent Peer Review file · Communications Chemistry]

Lid loop-mediated proton transfer revealed in the Fe/ α KG-dependent decarboxylase TraH

Corresponding Author: Dr Liujuan Zheng

Version 0:

Reviewer comments:

Reviewer #1

(Remarks to the Author)

The paper submitted by Zheng et al., have presented the X-ray structure of TraH enzyme along with the state-of-the-art computational methods. I believe this is of high standard and should be published subjected to some revision. They have provided a reasonable X-ray structure in complex with the ligand. Also, good mutagenesis work performed as well to validate the claims about the loop region. The paper also highlights the importance of disordered region such as random coils/loops can be altered to improved activity.

Abstract: The non-heme 2OG enzymes are well known and well characterised enzyme system. Please why authors say in the abstract they are poorly characterised need to more specific?

Introduction

The introduction is okay there is lot of information for a reader and there is a lot talked about different 2OG dependent enzymes, and it is confusing the reader, also reader must almost be an expert in 2OG enzyme family to read the article. Please may I suggest presenting the introduction more clearly and focus on IPNS family.

Please introduce DSBH in the introduction for the readers.

There is a great review on 2OG such as work done by Solomon's group etc.

Please provide a reference for this text 'Structural differences between these subfamilies, particularly unique insertions at the N- and C-termini of IPNS enzymes and the central region of TauD enzymes (Figure 1B), likely influence substrate recognition'

Methods:

Please provide all the scripts used for QM MM MD simulations using CP2K so that the keen readers might reproduce this work or apply this method to other systems.

How many replica runs were performed during the MD simulations ?

Results

Please provide the reference for the open and the closed state of the crustosic acid?

QM/MM results seem reasonable however authors have not reported any spin densities to validate their claims of the proposed reaction mechanism?

Please provide the radial distribution function for the water molecules presence near the Fe(IV)=O.

Also please provide some important distances especially of the Fe(IV)=O for the HAA from the MD trajectory.

Please provide the detailed view of the active site intermediates in Figure 5B, as the coordinating side chains of His, Asp, succinate are missing?

Please describe the coordination environment of Fe(IV)=O to be either 5C or 6C? Some time there might be a water molecule coordinating the six-coordination site. This will affect the barriers obtained.

Reviewer #2

(Remarks to the Author)

This manuscript presents investigation of TraH, a fungal iron/ α KG-dependent decarboxylase, and makes a contribution to our understanding of this enzyme family. The determination of multiple substrate-bound crystal structures of TraH provides a view of its active-site architecture, while the discovery of the N-terminal lid loop as a dynamic regulatory element is insightful. The authors' integration of structural biology, targeted mutagenesis, and advanced QM/MM simulations yields a mechanistic model in which a water-mediated proton relay, governed by the lid loop. The experimental design is rigorous and the conclusions are both well supported. It will be of broad interest to researchers in enzymology, structural biology, and natural-product biosynthesis. Some suggestions are included.

1. It is proposed that hydrogen-bond network is important for decarboxylation of crustosic acid. I wonder if the mutants with substitution of Q99, D112, and K114 can bind crustosic acid. It is possible that hydrogen-bond network is important for substrate binding and have small impact on decarboxylation.
2. Figure 5: Normal arrows should be used for proton transfer (not single arrow).
3. Figure 5: For the second cycle of the reaction, K191 after step 3 should be protonated again to regenerate the initial state. Please explain how this protonation happens.

Reviewer #3

(Remarks to the Author)

The study by Zheng et al. concerns the reaction mechanism of TraH, a Fe/ KG-dependant decarboxylase from *Penicillium crustosum* belonging to the IPNS family. They reported on the structural analysis of TraH, mutagenesis and QM/MM simulations. Their results show that a flexible N-terminal lid loop plays a crucial role in substrate recognition. Evolutionary studies suggest that this particular role has been conserved across the IPNS family.

This study builds on two previous papers (JACS 2019, OrgLett 2019) that reported the identification of the biosynthetic gene cluster of penilactones, as well as the identification and characterization of TraH, the decarboxylative enzyme catalyzing the first step in the conversion of crustosic acid to terrestric acid.

These results, although interesting, do not represent a significant advance in the fields of enzyme mechanism or structural analysis. Moreover, in my opinion, even if biochemistry is an adjacent field, the subject of the manuscript is too far from the core subject areas of Communications Chemistry. However, the work deserves to be published, and I would suggest submitting it to Scientific Reports. For future publications, I strongly recommend addressing several major points to substantially improve the quality and impact of the work. Major revisions are needed, particularly to clarify the scientific rationale and strengthen the experimental results. Addressing these aspects will significantly enhance the robustness and relevance of the manuscript.

The first question concerns the form of the substrate. P3, line 6 and P6, lines 3&20 + Fig3E: previous studies were mentioned without reference or support. P6, line 3 authors state that crustosic acid exists in two forms in equilibrium, which is an important point and this statement must be supported by experimental evidences. 1a and 1b have exact masses of 254.08 and 272.09 respectively; the decarboxylative product 2 has an exact mass of 208.07 while the open counterpart 2' has an exact mass of 226.08. If 1b is the true substrate of TraH, we would expect the decarboxylative product to be also in the open form or in equilibrium between the open and closed forms. All forms should be visible in chromatogram in Fig3E.

Ring opening needs water addition and opening of the resulting hemiacetal; it seems unlikely that this transformation occurs spontaneously in the biosynthetic pathway of terrestric acid, as the open forms 1b and 2' would have been observed by LC-MS in previous studies.

Mutations: what is the rationale for the mutations at Q99, D112, K114, K191 and Y214? The authors must comment on that and explain their choice. To eliminate H-interactions while maintaining hindrance, mutations of Q99 to L or I would have been appropriate. Q99R has weak activity, probably due to a preserved network of H bonds, while the other mutations, Q99E, Q99S and Q99K, suppress the network for various reasons, which should be discussed. While the choice for D112L mutation is understandable, being nonpolar with the similar hindrance, I question the mutations K114R, K191R and Y214F. What were the expected effects? Similarly, the E98Q and Q100E mutations in the loop do not appear to be relevant, or else this needs to be explained, as they have the same steric hindrance and remain polar.

Mechanism: in their 2019 paper (<http://dx.doi.org/10.1021/acs.orglett.9b04002>), the authors already proposed this mechanism, the only difference being the 1a closed form of the substrate. In their proposition, the key decarboxylation step occurs in a concerted fashion. Would a step wise mechanism through formation of a carbocation, in a similar way to PlsnB

(ref 8) be considered? Even if not benzylic as in PlsnB substrate, the potential carbocation would also be a stable intermediate.

Throughout the manuscript, the authors repeatedly draw a parallel between the fate of substrates 1 and 3—oxidative decarboxylation versus desaturation—as well as their different binding modes. This comparison, however, appears inappropriate, as it essentially contrasts two unrelated processes. Methyl ester 3 (in either its open or closed form) does not interact with the active site residues in the same manner as substrate 1 due to the presence of the methyl ester function. Lacking a carboxylate group, it cannot undergo oxidative decarboxylation, and the observation of an oxidative product is therefore not unexpected given that TraH is an oxidative enzyme. The authors may wish to reconsider or better justify this comparison, as it currently risks being misleading.

Lid loop role in substrate recognition. This section is of particular interest, and the authors may wish to discuss the work in parallel with the study conducted on the Clavaminase Synthase Like family, a sub-group of a-KG dependant enzymes, as the conclusions are similar (ref 18). This would further strengthen the manuscript and place the results in a broader context.

Minor comments

Fig1A is unclear

Correct Fig1A captions "bacteria" not "bacateria"

Fig1B : scheme EFE, to be corrected, ethylene + 3 CO₂ (ref 13)

P3, line 5, ref 2 not relevant, too specific; cite: Dinglasan, J.L.N., Otani, H., Doering, D.T. et al. Microbial secondary metabolites: advancements to accelerate discovery towards application. Nat Rev Microbiol 23, 338–354 (2025). <https://doi.org/10.1038/s41579-024-01141-y>

P4, line 26: better to show real alignment with IPNS structure, eg ScoE (PDB 6DCH)

P6, line 10: ref 20 refers to thymine hydroxylase, not substrate with tetronate core. The "clamps" are probably the consequence of pi-stacking interactions / VdW interactions

"residues F286 and Y214 make pi-pi stacking interactions with the conjugated enone of the tetronate core"

P6, line 10: G215 seems to be in indirect interaction with the substrate via water molecule, through H-bonds.

P6, line 5: correct the arrows in the proposed mechanism in Fig4B. Formation of double with single electron and one electron form C-H bond, the other one forming bond with OH to give H₂O. Single electron from Fe-OH to Fe(III) to be reduced to Fe(II).

P8, line 9, Fig S3A: the substrate must be drawn in the proposed mechanism.

P8, line 10: studies cited in references 8, 16 and 22 proposed mechanism where a residue in its protonated form or H₂O gave a proton. In TraH K191 and K114 are not under their basic form but protonated. This must be specified.

P9, line 2: to be corrected, K191 donating its proton, acts as a catalytic acid.

P11, line 5: did the authors used the predictive alphafold models from UniProt or did they perform the prediction themselves and if yes, the used tool must be specified.

P12, line 15: the proton is transferred to OH not to Fe centre.

Version 1:

Reviewer comments:

Reviewer #1

(Remarks to the Author)

I am happy with the correction made and this manuscript is ready for publication.

Reviewer #2

(Remarks to the Author)

The authors have responded to reviewers' concerns and the manuscript is now acceptable for publication.

Reviewer #3

(Remarks to the Author)

The authors have addressed the reviewers' comments and incorporated the suggested revisions; the paper has significantly improved as a result, particularly the discussion regarding the lid-loop significance. There are still a few minor corrections to be made before acceptance.

Fig4A, mechanism: correct the arrows from the radical intermediate to the final product.

The single-electron transfer to Fe(III) would generate a carbocation. The OH group from Fe(III)-OH would then abstract the proton, leading to the formation of a double bond.

One can envision that the mechanism proceeds in a concerted manner.

Fig5A, mechanism: in INT1, the single electron transfer occurs to Fe(III) core, not to the Fe(III)-OH bond; correct the arrows in INT2, they were correct in the previous version of the manuscript.

Fig7D: OH does not appear to be attached to the C but rather positioned to the side, please correct this.

Response letter

We thank all reviewers for their constructive and interesting comments, which we have mostly addressed. This significantly improved our paper

Reviewer #1 (Remarks to the Author):

The paper submitted by Zheng et al., have presented the X-ray structure of TraH enzyme along with the state-of-the-art computational methods. I believe this is of high standard and should be published subjected to some revision. They have provided a reasonable X-ray structure in complex with the ligand. Also, good mutagenesis work performed as well to validate the claims about the loop region. The paper also highlights the importance of disordered region such as random coils/loops can be altered to improved activity.

Abstract: The non-heme 2OG enzymes are well known and well characterised enzyme system. Please why authors say in the abstract they are poorly characterised need to more specific?

We appreciate the reviewer's overall positive comments and the questions. We have revised the Abstract to clarify that our study focuses specifically on the fungal IPNS-type decarboxylase, for which functional and mechanistic insights were previously lacking, thereby avoiding any misunderstanding that the entire 2OG enzyme family is poorly characterized

The introduction is okay there is lot of information for a reader and there is a lot talked about different 2OG dependent enzymes, and it is confusing the reader, also reader must almost be an expert in 2OG enzyme family to read the article. Please may I suggest presenting the introduction more clearly and focus on IPNS family.

Please introduce DSBH in the introduction for the readers.

We added the introduction of DSBH, as seen in Page 3, line 1-3. We have also revised the second paragraph of the introduction to better focus on IPNS-like family.

There is a great review on 2OG such as work done by Solomon's group etc.

We read the paper and add the sentences as well as cite the paper (Ref. 13).

Please provide a reference for this text 'Structural differences between these subfamilies, particularly unique insertions at the N- and C-termini of IPNS enzymes and the central region of TauD enzymes (Figure 1B), likely influence substrate recognition'

We thank the reviewer for this comment. In the revised manuscript, we have removed the sentence requesting a citation (and the surrounding speculative comparison) from the Introduction to better focus on IPNS-like family.

Methods: Please provide all the scripts used for QM MM MD simulations using CP2K so that the keen readers might reproduce this work or apply this method to other systems. How many replica runs were performed during the MD simulations?

Response: We thank the reviewer for this helpful suggestion. To ensure full reproducibility of our QM/MM MD and metadynamics simulations, we have now provided all relevant input files as Supplementary Data. Specifically, we include:

- (a) CP2K input files used for QM/MM MD simulations.
- (b) PLUMED input file used for the metadynamics simulations.
- (c) Topology and coordinate files required to construct the QM/MM system (Figure S4–S5).

These files enable interested readers to reproduce our simulations or adapt the workflow to other enzyme systems.

Regarding the MD simulations, we performed three independent replica runs for each system. This information has been clearly described in the Methods section of the revised manuscript.

Please provide the reference for the open and the closed state of the crustosic acid?

We have added the reference, please see at Page 5, line 22 (Ref.10).

QM/MM results seem reasonable however authors have not reported any spin densities to validate their claims of the proposed reaction mechanism?

Response: We thank the reviewer for this valuable comment. For Fe/ α KG-dependent enzymes such as TraH, the spin state of the ferryl intermediate is essential for establishing the correct reaction mechanism. We clarify in the revised manuscript that all QM/MM calculations were performed in the quintet state ($S = 2$), which is the well-established ground state for Fe(IV)=O intermediates in the Fe/ α KG enzyme family.

Furthermore, we have now emphasized that the spin densities for the Fe center, the oxo/hydroxo ligand, and the substrate radical were computed and are fully reported in Table S3. These results provide quantitative validation for the proposed HAT \rightarrow SET mechanism, where the substrate radical is confirmed as a discrete intermediate before being quenched by the Fe(III) center.

Please provide the radial distribution function for the water molecules presence near the Fe(IV)=O.

Response: We thank the reviewer for this suggestion. The radial distribution function (RDF) of water molecules around the Fe(IV)=O center in TraH has now been calculated and is presented in Figure S5E. This analysis confirms the presence of water near the reactive ferryl oxygen during the hydrogen atom abstraction step, providing additional support for the reaction mechanism described in the manuscript.

Also please provide some important distances especially of the Fe(IV)=O for the HAA from the MD trajectory. Please provide the detailed view of the active site intermediates in Figure 5B, as the coordinating side chains of His, Asp, succinate are missing? Please describe the coordination environment of Fe(IV)=O to be either 5C or 6C? Some time there might be a water molecule coordinating the six-coordination site. This will affect the barriers obtained.

Response: We thank the reviewer for these helpful suggestions. The key distance between Fe(IV)=O and the hydrogen atom on substrate 3b during the hydrogen atom abstraction (HAA) step has been extracted from the MD trajectories. Over the three independent replicas, the Fe=O-H_{3b} distance mainly fluctuates between 2.4–2.8 Å, with the average and standard error of the mean (SEM) indicated in Figure S5F.

The active site coordination environment of Fe(IV)=O has been clarified. Six-coordinated (6C) of Fe was used in the QM/MM based on the previous studies (ACS Catal. 2022, 12, 6, 3689–3699), bound to two histidines, one aspartate, succinate, and the oxo ligand, with a water molecule occasionally occupying the sixth site. A detailed view of the intermediates, including all coordinating side chains, has been added to Figure S4B. Nevertheless, we cannot rule out the possibility of a five-coordinate (5C) species. We thank the reviewer for raising this point and agree that it would be very interesting to investigate the potential differences between the 5C and 6C pathways in future studies.

These modifications ensure that readers can clearly visualize the active site geometry and understand the HAA distances used in the mechanistic analysis.

Reviewer #2 (Remarks to the Author):

This manuscript presents investigation of TraH, a fungal iron-dependent decarboxylase, and makes a contribution to our understanding of this enzyme family. The determination of multiple substrate-bound crystal structures of TraH provides a view of its active site architecture, while the discovery of the N-terminal lid loop as a dynamic regulatory element is insightful. The authors' integration of structural biology, targeted mutagenesis, and advanced QM/MM simulations yields a mechanistic model in which a water-mediated proton relay, governed by the lid loop. The experimental design is rigorous and the conclusions are both well supported. It will be of broad interest to researchers in enzymology, structural biology, and natural product biosynthesis. Some suggestions are included.

We sincerely thank the reviewer for the overall positive comments and the suggestions.

It is proposed that hydrogen-bond network is important for decarboxylation of crustosic acid. I wonder if the mutants with substitution of Q99, D112, and K114 can bind crustosic acid. It is possible that hydrogen-bond network is important for substrate binding and have small impact on

decarboxylation.

We thank the reviewer for this helpful suggestion. The key insight comes from comparing the two substrates. The methyl ester, which cannot use the hydrogen-bond network, shows similar binding and largely unaffected activity in our mutants. This indicates that the network is not required for general substrate engagement.

The complete loss of decarboxylation specifically with the carboxylic acid substrate suggests a more direct role in the chemistry itself. Our simulations support this by showing how the network facilitates proton transfer—a function that depends on precise substrate positioning but operates independently of initial binding affinity.

We agree that we cannot exclude mutations at Q99, D112, and K114 might impact the binding of crustosic acid. We have revised discussion (Page 13, para.2) to address this point explicitly.

Figure 5: Normal arrows should be used for proton transfer (not single arrow).

This has been corrected in the revised manuscript.

Figure 5: For the second cycle of the reaction, K191 after step 3 should be protonated again to regenerate the initial state. Please explain how this protonation happens.

Response: We thank the reviewer for this comment. After the third step of the catalytic cycle, K191 is indeed expected to be deprotonated to regenerate the initial state. Based on our MD simulations, this protonation is likely mediated by nearby solvent water molecules, which can serve as proton donors to the deprotonated lysine. While the exact proton transfer pathway was not explicitly simulated, the presence of stable water molecules near K191 (W1 and W2) suggests a feasible route for proton shuttling from the solvent to the lysine side chain.

Reviewer #3 (Remarks to the Author):

*The study by Zheng et al. concerns the reaction mechanism of TraH, a Fe²⁺/KG-dependant decarboxylase from *Penicillium crustosum* belonging to the IPNS family. They reported on the structural analysis of TraH, mutagenesis and QM/MM simulations. Their results show that a flexible N-terminal lid loop plays a crucial role in substrate recognition. Evolutionary studies suggest that this particular role has been conserved across the IPNS family.*

This study builds on two previous papers (JACS 2019, OrgLett 2019) that reported the identification of the biosynthetic gene cluster of penilactones, as well as the identification and characterization of TraH, the decarboxylative enzyme catalyzing the first step in the conversion of crustosic acid to terrestric acid.

These results, although interesting, do not represent a significant advance in the fields of enzyme mechanism or structural analysis. Moreover, in my opinion, even if biochemistry is an adjacent field, the subject of the manuscript is too far from the core subject areas of Communications Chemistry. However, the work deserves to be published, and I would suggest submitting it to Scientific Reports. For future publications, I strongly recommend addressing several major points to substantially improve the quality and impact of the work. Major revisions are needed, particularly to clarify the scientific rationale and strengthen the experimental results. Addressing these aspects will significantly enhance the robustness and relevance of the manuscript.

We sincerely thank the reviewer for reviewing our paper. We have substantially revised the manuscript based on the comments of the reviewer.

We believe these revisions now make the novel mechanistic insight and the broader relevance of our work to the chemistry community much clearer, and we thank the reviewer for prompting this important clarification.

The first question concerns the form of the substrate. P3, line 6 and P6, lines 3&20 + Fig3E: previous studies were mentioned without reference or support. P6, line 3 authors state that crustosic acid exists in two forms in equilibrium, which is an important point and this statement must be supported by experimental evidences. 1a and 1b have exact masses of 254.08 and 272.09 respectively; the decarboxylative product 2 an exact mass of 208.07 while the open counterpart 2' has an exact mass of 226.08. If 1b is the true substrate of TraH, we would expect the decarboxylative product to be also in the open form or in equilibrium between the open and closed forms. All forms should be visible in chromatogram in Fig3E.

Ring opening needs water addition and opening of the resulting hemiacetal; it seems unlikely that this transformation occurs spontaneously in the biosynthetic pathway of terrestrial acid, as the open forms 1b and 2' would have been observed by LC-MS in previous studies.

We thank the reviewer for the insightful comments. In this work, LC-MS monitoring of both substrate (**1**) and product (**2**) revealed merely the existence of their closed form (Fig S2), whereas crystallography only captured the open form (**1b**) bound in the TraH active site. This reflects that these compounds prefer the closed form in solution, while the open form is bound to the enzyme. The dynamic equilibrium between **1a** and **1b** has been demonstrated in our previous study (*Org. Lett.* 2020, 22, 88–92) by deuterium incorporation after incubation in D₂O.

Mutations: what is the rationale for the mutations at Q99, D112, K114, K191 and Y214? The authors must comment on that and explain their choice. To eliminate H-interactions while maintaining hindrance, mutations of Q99 to L or I would have been appropriate. Q99R has weak activity, probably due to a preserved network of H bonds, while the other mutations, Q99E, Q99S and Q99K, suppress the network for various reasons, which should be discussed. While the choice for D112L mutation is

understandable, being nonpolar with the similar hindrance, I question the mutations K114R, K191R and Y214F. What were the expected effects? Similarly, the E98Q and Q100E mutations in the loop do not appear to be relevant, or else this needs to be explained, as they have the same steric hindrance and remain polar.

We thank the reviewer for highlighting the need to clarify our mutagenesis rationale. To address this, we have added a new table (Table S2) detailing the structural basis and specific objective for each variant. The corresponding text in the Results section (Page 6, Para.2) has been revised to integrate this rationale, clearly linking each mutation's design to its experimental outcome, we believe these revisions provide the requested clarification.

In the meantime, we have also constructed Q99L as reviewer suggested. As expected, the mutant shows almost no decarboxylation activity to substrate **1**, but still show comparable desaturation activity to substrate **3** (Figure 3E and 4C), supporting our hypothesis for the function of the H-interactions.

Mechanism: in their 2019 paper (<http://dx.doi.org/10.1021/acs.orglett.9b04002>), the authors already proposed this mechanism, the only difference being the 1a closed form of the substrate. In their proposition, the key decarboxylation step occurs in a concerted fashion. Would a step wise mechanism through formation of a carbocation, in a similar way to PlsnB (ref 8) be considered? Even if not benzylic as in PlsnB substrate, the potential carbocation would also be a stable intermediate.

Regarding the possibility of a stepwise mechanism akin to PlsnB, we have now expanded the Discussion section (please see page13, Line19-25) to explicitly address this point. As detailed there, while a stabilized carbocation intermediate is plausible in PlsnB due to benzylic conjugation, our QM/MM metadynamics simulations for the TraH system favor a concerted decarboxylation-protonation pathway without a distinct carbocation minimum on the free-energy surface.

Since the key advancement in this manuscript being the structural and computational validation of that mechanism and its dependence on the specific lid-loop architecture. We agree that a highly asynchronous or "carbocation-like" transition state cannot be ruled out, which might be an interesting point for follow up studies.

Throughout the manuscript, the authors repeatedly draw a parallel between the fate of substrates 1 and 3—oxidative decarboxylation versus desaturation—as well as their different binding modes. This comparison, however, appears inappropriate, as it essentially contrasts two unrelated processes. Methyl ester 3 (in either its open or closed form) does not interact with the active site residues in the same manner as substrate 1 due to the presence of the methyl ester function. Lacking a carboxylate group, it cannot undergo oxidative decarboxylation, and the observation of an oxidative product is therefore not unexpected given that TraH is an oxidative enzyme. The authors may wish to reconsider or better justify this comparison, as it currently risks being misleading.

We thank the reviewer for this point, which has strengthened the logic of our mechanistic conclusion. We have revised the Discussion section (Page13, Para. 2) to better articulate our reasoning.

Lid loop role in substrate recognition. This section is of particular interest, and the authors may wish to discuss the work in parallel with the study conducted on the Clavamate Synthase Like family, a sub-group of a-KG dependant enzymes, as the conclusions are similar (ref 18). This would further strengthen the manuscript and place the results in a broader context.

We thank the reviewer for this excellent suggestion. We agree that comparing our findings with the well-studied Clavamate Synthase-Like (CSL) family provides a valuable broader context. In the revised manuscript, we have explicitly integrated this comparison into the discussion section (Page 13, lines 15-19).

Minor comments

Fig1A is unclear

Correct Fig1A captions "bacteria" not "bacateria"

Fig1B : scheme EFE, to be corrected, ethylene + 3 CO₂ (ref 13)

In line with the substantial revisions made to the Introduction section, the schematic has been updated to reflect the revised narrative. The pathway showing ethylene + 3 CO₂ is no longer included.

P3, line 5, ref 2 not relevant, too specific; cite: Dinglasan, J.L.N., Otani, H., Doering, D.T. et al. Microbial secondary metabolites: advancements to accelerate discovery towards application. Nat Rev Microbiol 23, 338–354 (2025). <https://doi.org/10.1038/s41579-024-01141-y>

We have replaced the previous reference at this position with the suggested review by Dinglasan et al. (2025). It is now cited as reference 2 at the indicated location (P3, line 3).

P4, line 26: better to show real alignment with IPNS structure, eg ScoE (PDB 6DCH)

We agree that a direct structural superposition would be insightful. However, due to significant structural divergence outside the conserved DBSH core domain (e.g., in peripheral helices and loops), a meaningful global alignment IPNS structure with TauD like enzyme is not feasible. As an alternative, we have provided a detailed sequence alignment (Fig. S9) that clearly highlights the conservation of key residues and, importantly, allows for the precise comparison of the lid loop regions from different subfamily.

P6, line 10: ref 20 refers to thymine hydroxylase, not substrate with tetronate core. The "clamps" are probably the consequence of pi -stacking interactions / VdW interactions

"residues F286 and Y214 make pi-pi stacking interactions with the conjugated enone of the tetronate core"

We have clarified the text to specify that residues F286 and Y214 mediate π - π stacking interactions with the tetronate core, effectively acting like "clamps" that stabilize substrate binding and positioning, analogous to the role of residues observed in homologous enzymes.

P6, line 10: G215 seems to be in indirect interaction with the substrate via water molecule, through H-bonds.

We revised the sentence to account for both direct and water-mediated interactions: "K191, H288, Y214, and G215 stabilize the substrate via hydrogen bonds, either directly or through water molecules."

P6, line 5: correct the arrows in the proposed mechanism in Fig4B. Formation of double with single electron and one electron form C-H bond, the other one forming bond with OH to give H₂O. Single electron from Fe-OH to Fe^{III} to be reduced to Fe^{II}.

The arrows in Figure 4B have been corrected to depict the electron flow accurately according to the suggestion.

P8, line 9, Fig S3A: the substrate must be drawn in the proposed mechanism.

we have revised Figure S3A to explicitly include the chemical structure of the substrate.

P8, line 10: studies cited in references 8, 16 and 22 proposed mechanism where a residue in its protonated form or H₂O gave a proton. In TraH K191 and K114 are not under their basic form but protonated. This must be specified.

We specified in the manuscript as following the sentences:

"For example, in SocE, a protonated R310 has been proposed to act as a proton donor during catalysis." and "Concurrently, protonated K191 in TraH, acting as a catalytic acid".

P9, line 2: to be corrected, K191 donating its proton, acts as a catalytic acid.

The text has been revised to indicate that K191 acts as a catalytic acid by donating its proton.

P11, line 5: did the authors used the predictive alphafold models from UniProt or did they perform the prediction themselves and if yes, the used tool must be specified.

We clarify that structural modeling of substrate-bound TraH homologs was performed using the Chai-I webserver, as described in the Methods section.

P12, line 15: the proton is transferred to OH not to Fe centre.

The discussion text has been revised to emphasize that K191 facilitates proton transfer to the iron-coordinated hydroxyl, without implying direct protonation of the metal itself.

Response letter

REVIEWERS' COMMENTS:

Reviewer #1 (Remarks to the Author):

I am happy with the correction made and this manuscript is ready for publication.

We greatly appreciate the reviewer's positive evaluation and helpful feedback.

Reviewer #2 (Remarks to the Author):

The authors have responded to reviewers' concerns and the manuscript is now acceptable for publication.

We sincerely thank the reviewer for the careful evaluation and helpful comments.

Reviewer #3 (Remarks to the Author):

The authors have addressed the reviewers' comments and incorporated the suggested revisions; the paper has significantly improved as a result, particularly the discussion regarding the lid-loop significance. There are still a few minor corrections to be made before acceptance.

We greatly appreciate the reviewer's positive evaluation and have addressed all minor corrections as suggested.

Fig4A, mechanism: correct the arrows from the radical intermediate to the final product. The single-electron transfer to Fe(III) would generate a carbocation. The OH group from Fe(III)-OH would then abstract the proton, leading to the formation of a double bond. One can envision that the mechanism proceeds in a concerted manner.

We have corrected the arrows

Fig5A, mechanism: in INT1, the single electron transfer occurs to Fe(III) core, not to the Fe(III)-OH bond; correct the arrows in INT2, they were correct in the previous version of the manuscript.

We have corrected the arrows

Fig7D: OH does not appear to be attached to the C but rather positioned to the side, please correct this.

We have changed the position now.